# Biotin proximity tagging favours unfolded proteins and enables the study of intrinsically disordered regions

David-Paul Minde [ID] [1,3]*, Manasa Ramakrishna [ID] [2,3] & Kathryn S. Lilley [ID] [1]*

Intrinsically Disordered Regions (IDRs) are enriched in disease-linked proteins known to have multiple post-translational modifications, but there is limited in vivo information about how locally unfolded protein regions contribute to biological functions. We reasoned that IDRs should be more accessible to targeted in vivo biotinylation than ordered protein regions, if they retain their flexibility in human cells. Indeed, we observed increased biotinylation density in predicted IDRs in several cellular compartments >20,000 biotin sites from four proximity proteomics studies. We show that in a biotin 'painting' time course experiment, biotinylation events in *Escherichia coli* ribosomes progress from unfolded and exposed regions at 10 s, to structured and less accessible regions after five minutes. We conclude that biotin proximity tagging favours sites of local disorder in proteins and suggest the possibility of using biotin painting as a method to gain unique insights into in vivo condition-dependent subcellular plasticity of proteins.

[1] Department of Biochemistry, Cambridge Centre for Proteomics, University of Cambridge, Tennis Court Road, Cambridge CB2 1QR, UK. [2] Medical Research Council Toxicology Unit, University of Cambridge, Lancaster Road, Leicester LE1 9HN, UK. [3] These authors contributed equally: David-Paul Minde, Manasa Ramakrishna. *email: dpm43@cam.ac.uk; k.s.lilley@bioc.cam.ac.uk

Cellular complexity often arises from structurally disordered proteins[1–4]. Intrinsically disordered regions (IDRs) within proteins often overlap with sites of alternative splicing and post-translational modifications (PTMs). Both splicing and PTMs together are estimated to expand the number of proteoforms into the millions despite a relatively compact (~20,000 large) protein-coding human genome[5–7]. Alternative splicing is frequent within the IDRs of proteins and can be a crucial element of PTM regulation, for instance by removal, recombination or modified local accessibility of potential sites of modification[3,8]. Proteins rich in IDRs, intrinsically disordered proteins (IDPs), are often linked to diseases, such as cancer, neurodegeneration and heart diseases[9–14]. Interest in understanding the role of IDPs is thus increasing within the biomedical research community.

Despite increasing community interest, it has remained challenging to define the phenomenon of intrinsic disorder as clearly as the ordered complement of the structural proteome. Rigidly folded proteins can be solved in high-resolution crystal, cryo-EM or NMR structures that can be described by a simplified hierarchy of elements of increasing length from primary structure (sequence of single amino acid) over secondary structure elements (α-helices and β strands of ~10 residues) to tertiary structure (folded domains of ~100 residues) and quaternary structures (i.e. assemblies of several folded proteins). IDPs cannot be as straightforwardly classified in a simple hierarchy of modules of increasing length because the minimal unit, a single IDR, can vary in length from a few residues to thousands of residues. Accordingly, IDRs can vary significantly in their properties and functions and the need for further differentiation of sub-classes of disorder was recognized early in the development of the field[15]. While the structure-function paradigm is fully established and has been highly successful, a complementary disorder-function paradigm is still emerging[16].

Co-evolutionary inference suggests that many predicted disordered regions have the capacity to fold and are selected in evolution by contact constraints imposed by the folded conformation in the presence of cellular binding partners[17]. In other words, such binding-coupled folding IDPs look similar to folded proteins as determined by (co)evolution statistical analysis. Interfaces of foldable IDRs tend to be larger than contacts between two ordered proteins and the exposed hydrophobic surface area is often larger, which in some cases limits the solubility of IDPs and requires tighter subcellular regulation of IDPs compared to ordered proteins[18–20].

One of the least characterized aspects in IDR research is in vivo malleability leading to multiple structural forms that disordered regions can adopt in a given compartment in a given cellular state. According to in vitro experiments, it can be expected that subtle variations in pH, salt concentrations, and PTMs can have very significant effects on the conformational ensembles of IDPs[21]. For instance, nuclear pore proteins can form extremely tight complexes (dissociation constant ($K_d$) in low pM range) near physiological salt concentration (~100 mM) which becomes very weak ($K_d$ in mM range) at 200 mM salt concentration[22]. Indeed, a large-scale multidimensional proteomics study that investigated temperature-dependent solubility and abundance changes across cell cycle phases, demonstrated that large subsets of the human proteome dramatically change their solubility, stability, subcellular organization and protein partners in patterns resembling differential phosphorylation during the cell cycle[23]. A more direct link between phosphorylation and stability changes has been discovered using a recent method termed 'hotspot thermal profiling' that combines thermal protein solubility and subsequent phospho-enrichment to quantify how 'phosphomodiforms' differ in their melting properties. This study showed that phosphorylation in loop regions induces the largest range of changes in the melting points of their respective proteins[24].

Early reports suggested that site identification in phosphorylation predictions can become significantly more accurate if local intrinsic disorder tendency is taken into consideration[25]. Many single-protein examples illustrate that IDRs can be phosphorylated or hyper-phosphorylated within disordered residues, often at highly soluble and intrinsic disorder-promoting serine and threonine residues[11,12,26–28]. The correlations of IDRs with acetylation, ubiquitination and sumoylation of lysine residues, and phosphorylation events at residues such as tyrosine and histidine are more challenging to detect, and hence are frequently under-reported in scientific studies[29–33]. Finally, there are very few studies reporting possible interactions between IDRs and multiple types of PTMs.

Biotinylation-based proximity proteomics methods are traditionally used to map transient interactions and subcellular neighbours[34–38]. The common principle of various proximity proteomics approaches is that biotinylation is highest in the vicinity of the biotin-activating enzyme that is fused to protein of interest. Most proximity proteomics studies aim to quantify biotinylated proteins in the vicinity of the biotin-activating enzyme. Because no direct sites of biotinylation are obtained in this approach, stringent statistical tests are required to remove endogenously biotinylated and non-specific avidin-bead binding proteins. Several recent technological improvements have enabled the direct detection of thousands of biotin sites in hundreds of proteins in a single study[39–42]. We, therefore, reasoned that these novel large-scale in vivo biotin site data could be repurposed to gain insights into possible cellular conformations of proteins.

The most frequently used enzymes in proximity proteomics are variants of BirA biotin-protein ligase and ascorbate peroxidase (APX)[36,37,43,44]. A promiscuous mutant of BirA (BioID) as well as a thermophilic homologue (BioID2) biotinylate nearby lysine residues through the formation of activated biotinoyl-5′-AMP which forms a covalent attachment to the nucleophilic ε-amino side chain group of lysine (K). APX or accelerated versions like APEX2 can convert biotin-phenol to activated radicals that can readily react for a short period of time with nearby tyrosine residues (Y). Interestingly, these two amino acid types are on opposite ends of the disorder-promoting amino acid scale—lysine promotes disorder while tyrosine is on average depleted in IDRs[45].

Here, we hypothesize that proximity labelling studies can be biased by structural features of target proteins and test this hypothesis using large-scale datasets. We demonstrate the enrichment of cellular biotinylation events in predicted IDRs of proteins in HEK293 cells which points to the re-purposing of in vivo biotinylation to achieve comprehensive conformational proteomics studies in intact cells.

## Results

**Concept of the study.** To gain insights into conformational plasticity of predicted IDRs in vivo, we explored whether subcellular biotinylation patterns can vary with the extent of predicted local flexibility of proteins (Fig. 1a). We reasoned that IDRs that remain unfolded and unconstrained by interactions in vivo would be expected to be more readily biotinylated given their large accessible surface and high dynamics[22,46]. Alternatively, IDRs that completely fold upon forming interactions with other proteins[47] should have average levels of biotinylation. Furthermore, IDRs that are comprised of concatenated short interaction motifs may simultaneously interact with multiple partner proteins that limit tagging efficiency by shielding interacting surfaces[48].

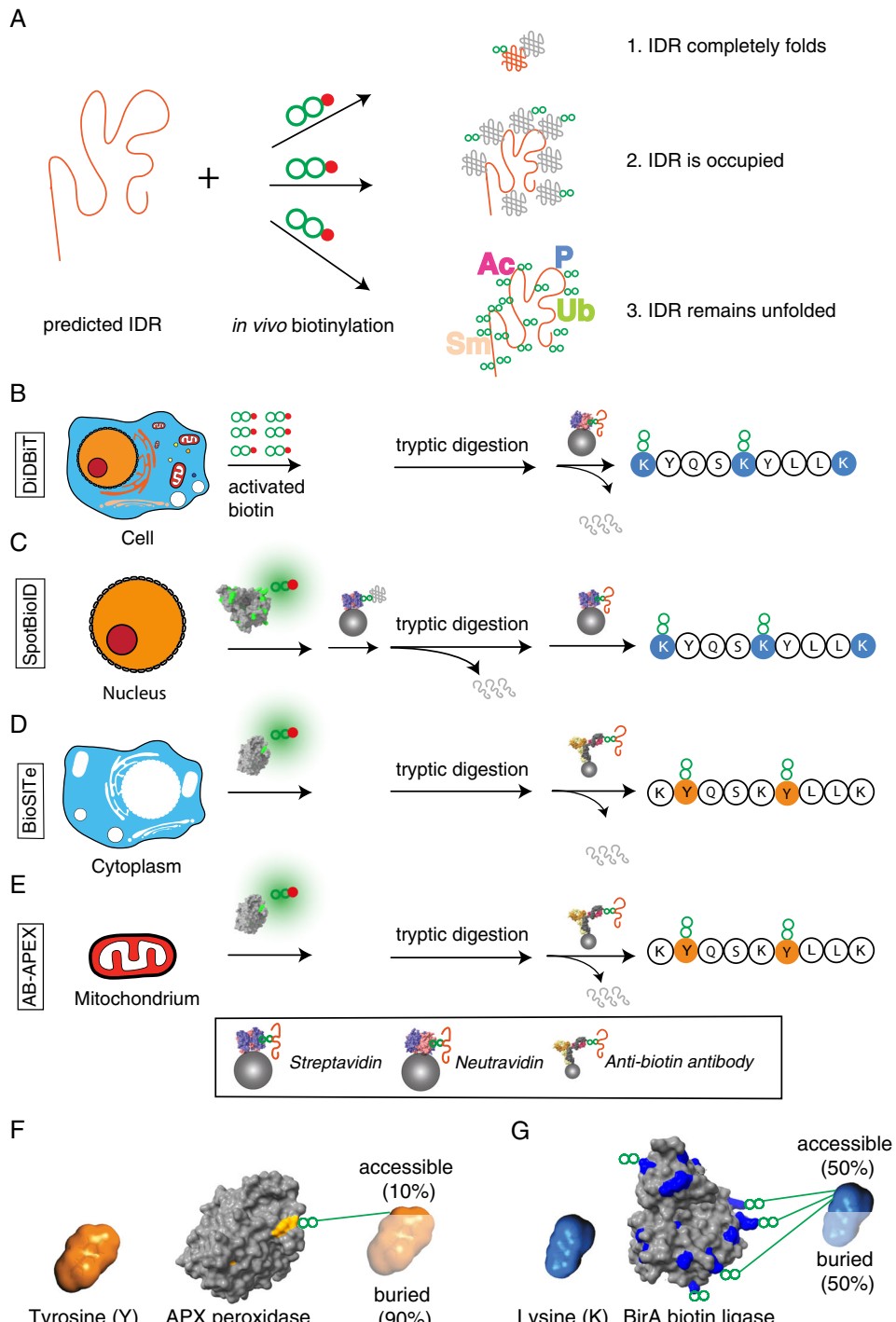

**Fig. 1 Large-scale in vivo biotinylation datasets can be re-purposed to identify accessible protein regions in vivo. a** Concept of the study. Predicted IDRs are compared with in vivo biotinylation sites and the most frequently reported post-translational modification sites to identify highly accessible regions in proteins. Study design of re-analyzed studies: **b** DiDBiT[41] **c** SpotBioID[42] **d** BioSITe[39] **e** AB-APEX2[40]. **f** Tyrosine solvent accessible surface area (SASA) is reduced significantly (~90%[50]) in folded proteins as illustrated on the APX structure (PDB ID 1APX). **g** Lysine sidechains contribute a large fraction of the total surface in typical folded proteins as illustrated in the Aquifex aeolicus BirA (BioID2) structure (PDB ID 3EFR). Some 50% of the average lysine's SASA stays exposed in folded proteins[50].

**Selection of orthogonal large-scale proximity proteomics studies to test our hypothesis**. To test our hypothesis of possible links between structural features of proteins and biotinylation, we selected four recent, independent, large-scale in vivo biotinylation studies by the following criteria: large number of directly identified biotinylation sites; orthogonality in targeted subcellular niches; and independence of biotin-peptide enrichment strategies (Fig. 1b–e).

Firstly, the DiDBiT study of Schiapparelli et al. targeted the whole cell and is therefore agnostic of subcellular localisation. This study identified ~20,000 biotinylation sites on lysine sites upon extensive biotinylation by applying 1 mM NHS-biotin, a chemically activated form of biotin, to cultured HEK293 cells, complete digestion by trypsin and streptavidin-affinity purification of biotinylated peptides[41]. Secondly, the SpotBioID study of

Lee et al. targeted rapamycin-dependent interactions of the human mTOR kinase using its FK506-rapamycin binding (FRB) domain fused to BioID[42]. Immunofluorescence data within the SpotBioID study conflicts with previous literature concerning the main subcellular localisation of FRB-BioID fusion that appears to be cytoplasmic in fluorescence experiments and nuclear in previous literature and biotin-protein enrichments[42], with most evidence suggesting a mainly nuclear localisation of the FRB-BioID fusion. The remaining two datasets come from recent, tyrosine-targeting APEX2 studies. Both successfully explored an alternative enrichment strategy based on polyclonal biotin-antibodies to achieve gentle elution while retaining explicit biotin site information unlike other strategies involving gentle elution of cleavable biotin derivatives[39,40,49]. They comprise an antibody-based APEX2 study (within this paper termed Ab-APEX) targeted the mitochondrial matrix using mito-APEX2[40], and a study called BioSITe[39] which uses a cytoplasmic APEX2 fusion construct to Nestin (NES) protein

## Orthogonality of tyrosine and lysine as molecular targets of proximity proteomics.

How different are tyrosine and lysine residues, the most frequent molecular targets in proximity proteomics? Tyrosine is a partly hydrophobic and bulky amino acid and predominantly partitions to the hydrophobic core of proteins and near the interface of intrinsic membrane proteins. Its solvent accessible surface area (SASA) shrinks by some 90% during folding reactions (Fig. 1f)[50]. Lysine residues, by contrast, tend to orient to the surface of folded proteins and stay in contact with surrounding water molecules, i.e. retain a large fraction of their SASA (Fig. 1g). Nevertheless, through their intramolecular and intermolecular contacts, for instance, in protein–protein interactions, lysine residues have a large spectrum of accessibilities with an average near 50% of remaining SASA in folded proteins[50]. We reasoned that the orthogonality of these two amino acids would be helpful to comprehensively test our hypothesis that in vivo biotinylation can be significantly favoured in predicted IDRs.

## Proximity proteomics studies can specifically target subcellular locations.

As expected, the four studies targeted different subcellular niches and thus there was a very small overlap in proteins across the four studies with only 29 proteins being in common (Supplementary Fig. 1a). Of these 29, many of them had multiple cellular locations predominating in the nucleus (Supplementary Fig. 1b, blue), cytosol (Supplementary Fig. 1b, red) and the extracellular region (Supplementary Fig. 1b, yellow). Given the small size of this subset of the whole dataset, these locations were not statistically enriched despite being frequently seen. However, we could confirm the location for each of the studies above ($n >$ 500) using a functional enrichment analysis against a set of gene ontology (GO) terms aimed at describing cellular location (GO: CC; Supplementary Fig. 1c). Our data analysis show that as expected, Ab-APEX proteins strongly target the mitochondrion with high fold enrichment for the mitochondrial matrix and the mitochondrial inner membrane (Supplementary Fig. 1c, first column). We then interrogated the BioSITe data which also as presumed based on the NES-APEX fusion, was enriched in GO terms of the cytoplasm and the cytosol (Supplementary Fig. 1c, second column). The DiDBiT study, which lacks specific targets appeared to be enriched for nuclear, mitochondrial and cytosolic proteins (Supplementary Fig. 1c, column 3). Finally, data from the SpotBioID study, where the authors state that FRB-BioID is cytoplasmic, were enriched for mostly nuclear and some cytoskeletal proteins[42].

We also explored the overlap in the subcellular assignment for these repeatedly identified proteins according to previous large-scale studies. We tested the consensus of assignments among efforts to map the subcellular distribution of proteins in the primarily immunofluorescence-based subcellular Human Proteome Atlas (HPA) study[51], a recent mass spectrometry-based 'Localisation of Organelle Proteins by Isotope Tagging after Differential ultracentrifugation' (LOPIT-DC) study[52] as well as the Uniprot database[53]. This analysis revealed that out of all repeatedly identified proteins across the four studies only one, namely the well-characterised endogenously biotinylated pyruvate carboxylase, showed a unique consensus location in the mitochondrion (Supplementary Data file 2). Given the different resolution of these different sources of subcellular location information, we also analysed if there is a partial consensus, for instance within different parts of the nucleus. We identified two examples of proteins that share a partial consensus location in the nucleus. Both proteins (Heterogeneous nuclear ribonucleoprotein M and TATA-binding protein-associated factor 2N) are predicted >70% disordered and share multiple sites of biotinylation across studies with the same target amino acid (Supplementary Data file 2). We also examined the patterns of biotinylation in the endogenously biotinylated mitochondrial pyruvate carboxylase. As can be reasonably expected, this enzyme showed more sites of biotinylation in the Ab-APEX study than in other studies that did not target the mitochondrion (Supplementary Data file 2). Briefly, all studies except SpotBioID showed expected enrichments consistent with the targeted cellular compartment and most previous literature.

## Illustrative examples of proteins that are biotinylated across all four independent studies.

We next explored the structural features of the limited subset of 29 proteins that were common in all studies by combining all biotinylation sites in the four datasets. While not statistically significant, we noticed that the list contained many RNA binding proteins. Elevated IDR content among these proteins is consistent with previous reports of high IDR content among nucleotide-binding proteins[54] but a larger set will have to be explored for firmly establish a statistical correlation. The first example, Emerin, is an integral membrane protein that is often found at the inner nuclear membrane or at adherens junctions. Emerin mutations cause X-linked recessive Emery–Dreifuss muscular dystrophy. Biotinylation sites from all four studies cluster in a large predicted IDR in the first half of the protein sequence, avoid the transmembrane-spanning domain (Fig. 2a–e) consistent with our hypothesis that predicted IDRs might be more biotinylated in vivo if they remain highly accessible. A very large number of other PTMs in this IDR further illustrates that this membrane protein is indeed often subjected to intracellular modifications (Fig. 2a, Supplementary Data file 1, BioPTM-IDR-correlation). Surprisingly, Emerin is found in all four studies despite the fact that some targeted different subcellular locations.

Emerin is the only integral membrane proteins that was repeatedly identified across all studies. An additional ~400 integral membrane proteins are identified in at least one of the four studies suggesting that detailed intracellular structural insights can be gleaned from re-purposed proximity proteomics studies.

Next, we analysed the predicted fully disordered RNA-interacting plasminogen activator inhibitor protein SERBP1 (Fig. 2f). Four sites of biotinylation, across the four studies, cluster around the central region of this protein (residues 200–260) where previously reported unique PTM sites also cluster (Fig. 2g). DiDBiT identifies many additional sites scattered

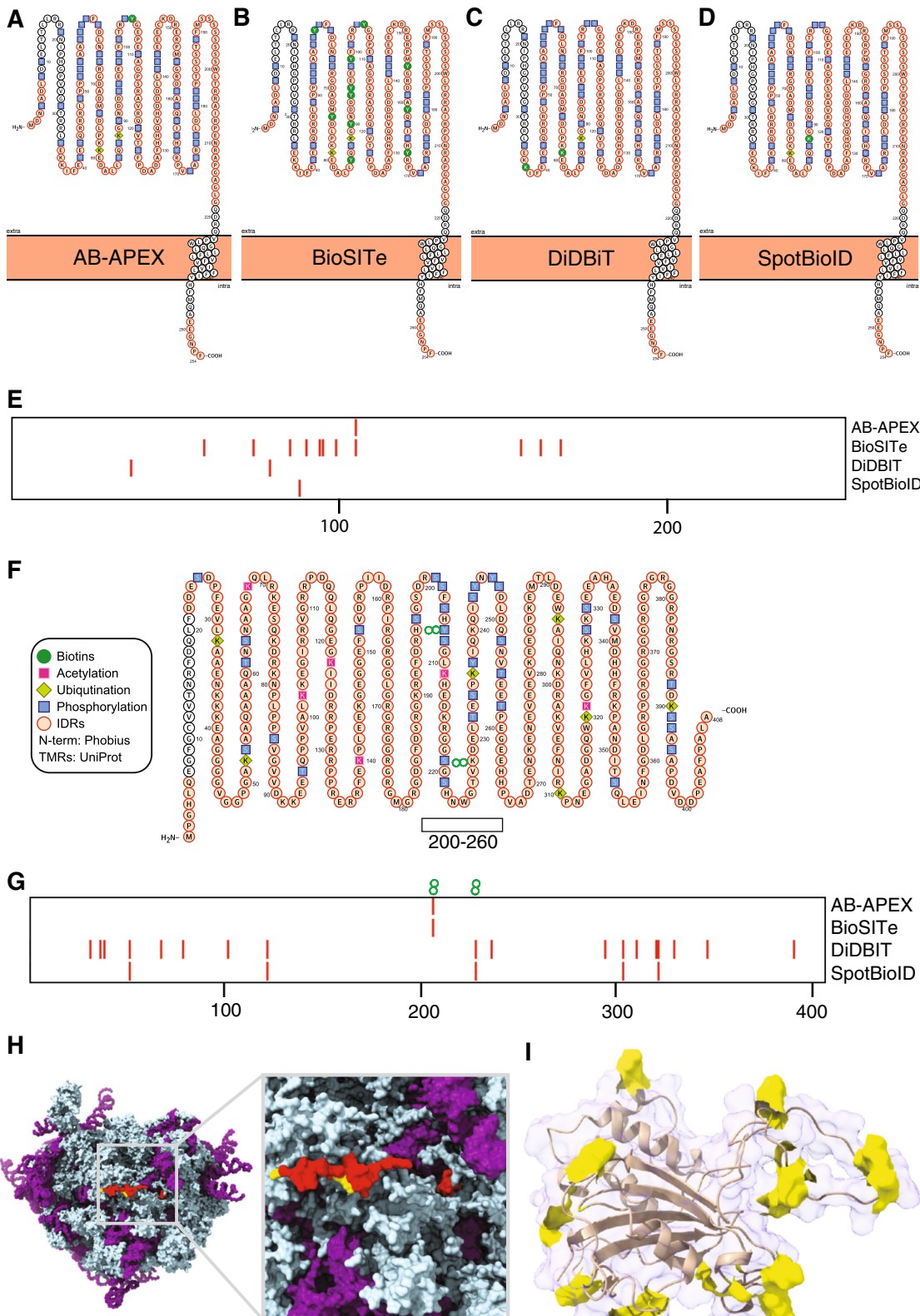

**Fig. 2 Illustrative examples for in vivo surface biotinylation in four independent studies.** Emerin: **a–d** Protter[55] representations of regions of IDR (orange), frequent PTMs and biotin modification in the four studies **e** comparison of biotinylation sites across four studies.SERBP1: **f** Protter plot showing post-translational modifications and regions of IDR (orange) **g** sites of biotinylation across four studies **h** Cryo-EM model (PDB ID 4V6X[56]) of SERBP1 with sites of biotinylation across all studies highlighted in yellow and the cryo-EM resolved fraction (~20%, most coil and α-helix) of SERBP1 highlighted in red. (RNA is denoted in purple) **i** FKPB3 NMR structure (PDB ID 2mph) with biotinylations in yellow, non-biotinylated chains are represented in cartoon style under 90% transparent surface. Note that the legend in **f** also applies to (**a–d**).

over the entire protein sequence, five of which are common with the nuclear targeted SpotBioID study. SERBP1 was previously found in multiple subcellular locations consistent with its identification in four studies. SERBP1 has a solved structure in which it is attaching at the periphery of the 80S ribosome RNA–protein complex and mostly lacks (throughout ~80% of its sequence) unique electron density (Fig. 2h); remaining small visible fractions form elongated structures that are detected in random coil or α-helical conformations. This SERB1 example provides anecdotal evidence that disordered regions can be densely biotinylated.

Finally, we selected FK506- and rapamycin-binding protein (FKBP3) as a protein of average (predicted) disorder content for the human proteome around 40% according to VSL2b[2]. FKBP3 is a cis-trans prolyl isomerase that is involved in cellular protein folding and tightly binds to the immunosuppressant rapamycin. We noted that biotinylation events were enriched in its predicted IDRs (72% of sites of biotinylation are within IDRs) or localised to local coil structure and short, highly accessible α-helical segments in the NMR structure (Fig. 2i).

We conclude that detailed inspection of common examples across four studies suggests an enrichment of biotinylations in IDRs and regions lacking defined secondary structure in otherwise folded proteins. Furthermore, the observed differences in biotinylation sites between studies is likely due to multi-localised proteins changing conformation and/or protein–protein interactions in different subcellular niches.

**Predicted IDRs are more frequently and densely biotinylated in vivo.** Encouraged by our observations that biotinylation events were enriched within the predicted IDRs in the small pool of proteins common to all four studies, we next considered whether this trend might still hold globally for the biotinylated proteome (referred to as biotinome hereafter) comprising nearly 4000 proteins. We first checked if proteins with higher predicted fraction of IDRs contain higher numbers of unique sites of biotinylation by comparing the predicted IDR fraction for proteins in each biotinome to the number of biotinylated sites they contained (Fig. 3a). Within each dataset, there were only a small number of proteins with five or more biotinylation sites and hence these have been collectively binned into the 5+ category (Fig. 3a, most right hand violin). For both the SpotBioID and the BioSITe studies, we observed an increase in the frequency of sites of in vivo biotinylations per protein from 1 to 4 with increasing IDR fractions, while DiDBiT and Ab-APEX did not show this trend (Fig. 3a). Both the cytosol and the nucleus, which are target compartments in BioSITe and SpotBioID have been previously suggested to contain many IDRs[54]. Mitochondria, by contrast, are predicted low in IDRs especially their subset of proteins with bacterial homologues[55]. The DiDBiT study, lacking compartmental preference, contains both highly disordered and fully folded proteins which might mask any possible weak correlation. Alternatively, it is possible that the long exposure of cells to very high concentrations of activated biotin in the DiDBiT study can saturate the less accessible and mostly folded regions in proteins. This may result from dynamic folding states of such proteins over time that are not captured with shorter labelling times and the limited concentrations of activated biotin in the vicinity of APEX2 or BirA variants in typical proximity studies. We concluded that in vivo observed biotinylation frequency per protein and predicted IDR fractions can be correlated in IDR-rich compartments such as the nucleus and cytosol in HEK293 cells (Supplementary Data file 1, Biotin-list).

To overcome limitations of averaging over IDRs and ordered regions that might have masked structural trends in the DiDBiT

and Ab-APEX studies, we next refined our analysis by distinguishing between biotinylation events inside and outside of IDRs while accounting for the density of potentially modifiable residues. To establish an expected rate of biotins, we calculated the number of lysine residues (K; for SpotBioID and DiDBiT) or tyrosine residues (Y; Ab-APEX and BioSITe)—both within the predicted regions of IDR (as determined by VSL2b) and across the entire protein body. The ratio of all K/Y residues within IDR regions to all K/Y residues across the protein body gave us an expected rate of biotinylation in IDRs. We then performed a similar calculation using the numbers of biotins we actually observed within IDRs and across the whole protein for each of our 4 studies (Fig. 3b). Other than in the DiDBIT study, we observed a significantly greater number of biotins within IDR regions (orange bars; Fig. 3b) than expected (blue bars; Fig. 3b) by one or more prediction algorithms. This observation was more significant in the cytoplasmic (BioSITe) and nuclear proteins (SpotBioID) than in the mitochondrial proteins (Ab-APEX) (Supplementary Fig. 2a).

Convinced that we are seeing a true positive correlation in three out of four studies, between local predicted IDRs and biotinylation density, we next sought to see if similar trends can also be observed on protein level after sorting all proteins in classes ranging from most to least folded. To this end, we labelled a protein as Folded (F) if it had predictions of <10% IDR, Partially Folded (P) if it had 10–30% IDR and Unfolded (U) if it had >30% IDR in its protein body similar to a strategy in Gsponer et al.[19]. We then looked at the overall distribution of proteins in these IDR classes for each of our 4 studies (Fig. 3c) where we display the results for just VSL2b and IUPred-L algorithms as VSL2b_IUPred-L mimics the trend of VSL2b alone while the D2P2 consensus mimics IUPred-L. We observed that all studies contain proteins that can be classified as F, P and U thus enabling pairwise comparisons. The predictors that are better at predicting long IDRs or the absence of folded domains, IUPred-L and D2P2 consensus predictors[56,57], classified more proteins as F than VSL2b that has a wider definition of IDRs that also includes short IDRs. Consistent with our previous observations and claims in literature[22,54], the nuclear protein enriched SpotBioID dataset shows the highest proportion of U proteins while the mitochondria targeting Ab-APEX study shows the highest proportion of F proteins (Fig. 3c, Supplementary Figs. 2b, 3).

Given these three categories of proteins, we wondered whether there would be an association between IDR-associated-biotins and the various categories of IDPs. To assess this, we performed both pairwise t-tests between the groups (F-P, U-P, U-F; Supplementary Fig. 2c) and an ANOVA across all groups followed by a Tukey's Honestly Significant Differences post-hoc test (Supplementary Fig. 2d). In all studies except the SpotBioID study, there were significant differences between biotin numbers in the F and U group with more biotinylation events occurring in the U group. Additionally, the differences were significant for all studies between U and P groups, once again showing higher number of biotins in the U group (Fig. 3d; Supplementary Fig. 2c).

To investigate whether larger structured complexes can also be analysed with biotin 'painting', we filtered the DiDBiT dataset for ribosomal proteins and visualised all biotinylated subunits in an exploded version of the 80S ribosome (Fig. 4 and Supplementary Fig. 4). Virtually all biotinylated subunits are non-globular and contain many biotinylations. Many of these are inaccessible to water or larger molecules such as biotin in the fully assembled 80S ribosomal complex as they are contacting ribosomal RNA (Supplementary video 1). High biotinylation density in the 80S ribosome is consistent with an earlier suggestion that eukaryotic ribosomes are rich in predicted IDRs that can be functionally

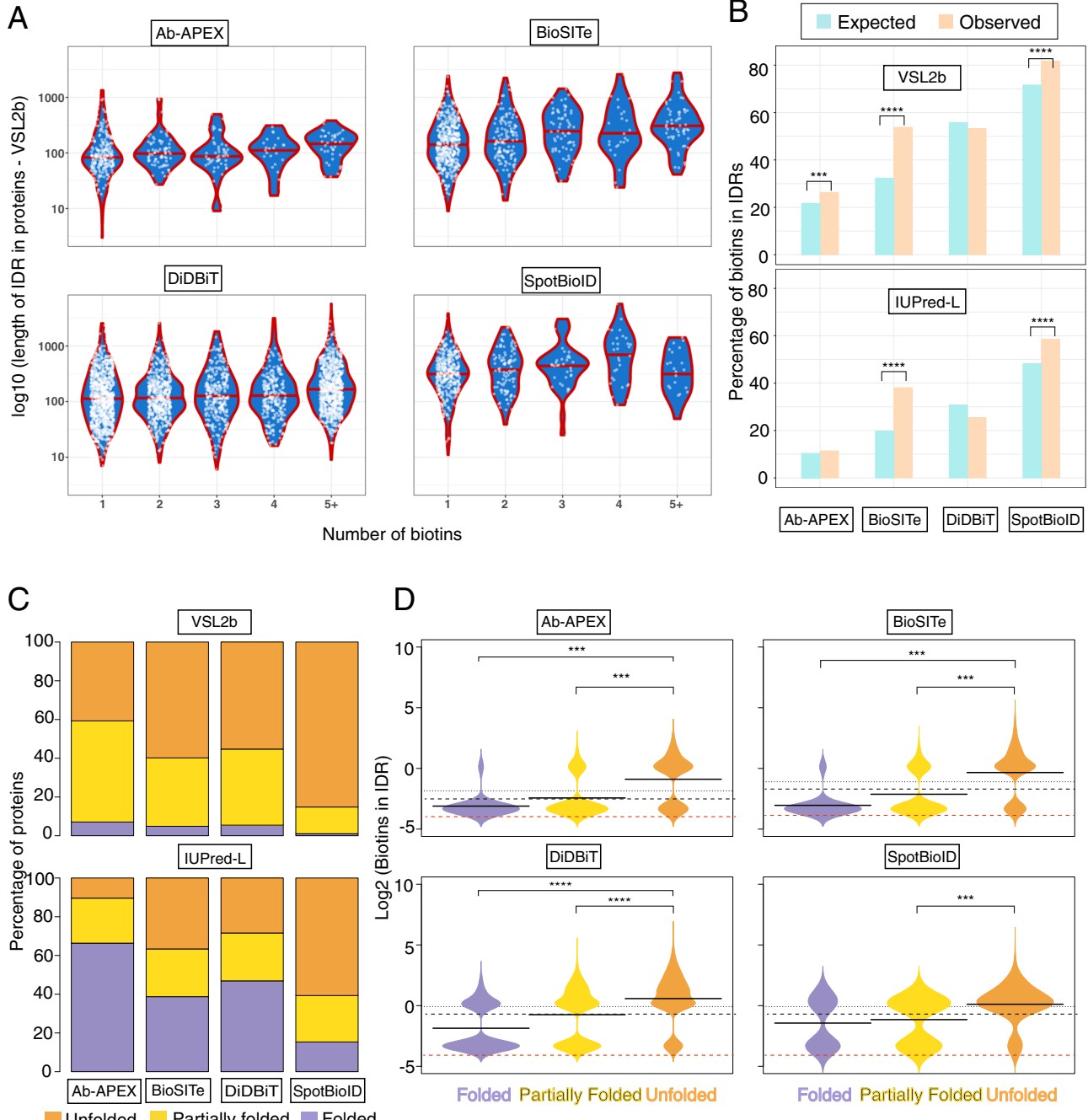

**Fig. 3 Predicted IDRs are preferentially in vivo biotinylated across all studies a** Violin plots (i.e. mirrored density distribution plots) showing the relationship between number of biotins and length of IDR across all biotinylated proteins. Biotin numbers > = 5 are grouped into one set to show the general trend of the data. The red line in the middle of each violin represents the median fraction of IDR for that group. **b** Barplots showing the Expected (pale blue) and Observed (pale orange) distribution of biotins within regions of IDR across the 4 studies using the IDR caller VSL2b (Top) and IUPred-L (bottom). Significance: ***$p < 0.0005$; ****$p$ close to 0, using a binomial test where the "probability of success" is the (number of lysine residues or tyrosine residues in IDRs/Total number of lysine residues or tyrosine residues), a "success" is a biotin within an IDR and "number of trials" is the number of biotins observed in that study. **c** Barplots showing the distribution of proteins from the four studies across the three structural classes[19]: Folded (F, 0–10% disorder; purple); Partially Folded (P, 10–30% disorder; Yellow) and Unfolded (U, >30% disorder; orange) for two different IDR callers VSL2b (Top) and IUPRed-L (bottom). The numbers of proteins in the VSL2b caller are displayed in Supplementary Fig. 2b. We note that disorder classifications vary substantially with specific predictors, which is expected given their differences in precision and recall. At least for SpotBioID, all tested prediction tools indicate that most sites of biotinylation map to partially folded or unfolded proteins. **d** Bean-plots[58] showing the distribution of biotins that occur within VSL2b predicted IDRs across the 3 classes F, P, U in each study. The y-axis in on a log2 scale with values 0 and above representing 1 or more biotins. The red dotted line represents 0 biotins (with added correction factor). The solid black line in the middle of each violin represents the mean biotins (on log2 scale) for that group. The black dotted line represents the mean log2(Biotins) across all groups.

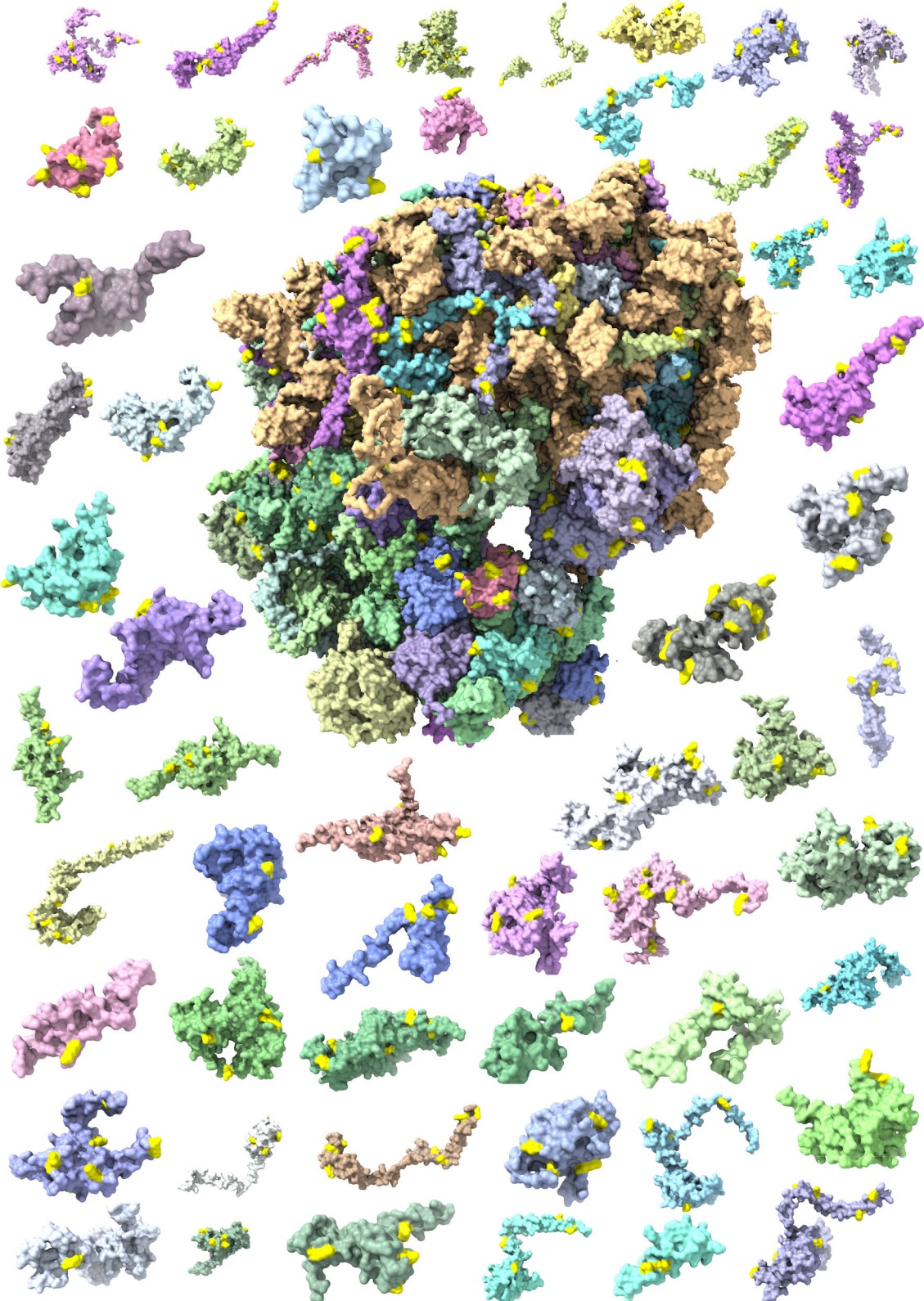

**Fig. 4 Sites of in vivo biotinylations mapped on in silico disassembled 80S ribosome (PDB: 6EK0).** An exploded ribosome plot showing the individual proteins that make up the eukaryotic 80S ribosome with added biotinylation marks (bright yellow) from four biotinylation datasets. We can see that nearly all ribosomal proteins have some yellow "paint" on them.

essential. Additionally, it could indicate that elongated shapes of ribosomal proteins that show unusually large interfaces are likely to have low intrinsic stabilities upon dissociation from partner proteins and ribosomal RNA[58,59]. Our statistical tests indicated significant enrichment of biotinylation density in ribosomal IDRs beyond trivially expected modification levels based on the number of available residues that can be modified by activated biotin (Supplementary Data file 1, Ribo-stats). We also extended this analysis to the mitoribosome and found a similar enrichment of biotinylation in IDRs of the mitoribosome (Supplementary Data file 1, Ribo-stats). We observed frequent colocalization of sites of biotinylation in vivo and high B-factors as measure of local mobility according to high-resolution structural cryo-EM analysis in vitro (PDB: 3j9m[60], Supplementary Fig. 5).

**Limited biotin painting maps to mobile regions within the bacterial ribosome.** To test empirically how fast both unstructured and structured regions in ribosomes can be biotinylated, we performed a multiplexed proteomics experiment that explored the time-dependence of biotinylation in purified 70S ribosomes from *E. coli*. We labelled purified 70S ribosomal complexes using 1 mM NHS-biotin for either 10 s, 120 s or 300 s and subsequently labelled these peptides with different TMT tags to quantify the relative intensity of biotinylation in these different timepoints for all detectable sites of biotinylation. We observed the highest preference for disordered regions at short labelling times (10 s) and deeper structural penetration after five minutes of labelling (Supplementary Fig. 6, Supplementary Fig. 7, Supplementary Data file 3). Clearly, biotin painting was more limited to highly exposed and disordered regions at limiting times of incubation with activated biotin.

## Discussion

We describe here the first in vivo evidence for preferential biotinylation of predicted IDRs in four independent proximity proteomics studies, to our knowledge. This observation adds a new type of (exogenous) tag to a list of naturally occurring PTMs (phosphorylation, sumoylation) that have previously been suggested to be enriched in IDRs[25,33] and which we have further confirmed within this dataset (Supplementary Data file 1, Supplementary Fig. 8). This raises an important question as to whether proteins first unfold in vivo before modification or unfold after modifications. It will be interesting to comprehensively explore the proteome-wide interplay between structure and the rate of addition and removal of specific PTMs. Similar to Hotspot Thermal Profiling[24], which compares thermal solubility of proteomes and their phosphorylated complement to detect the impact of phosphorylation on thermal stability of specific proteins, biotin painting might be combined with enrichment of specific PTMs to probe the local structural impact of these PTMs. If specific PTMs favour local unfolding, one would expect more biotinylation in their vicinity upon enrichment. Alternatively, some PTMs might induce local folding or enhance binding affinity with partner proteins and thereby limit local accessibility. Biotin painting shows promise to provide new global insights into PTM-induced structural rewiring in cells.

Ribosomal proteins piqued our particular interest both because of their high cellular abundance and because they help to accelerate the most diverse biosynthesis of nearly all polypeptides and proteins, which requires exceptional flexibility and functional plasticity. We observed that many of ribosomal proteins had biotinylation marks in regions that are covered by RNA in structural snapshots of the 80S particle (Fig. 4 and Supplementary Fig. 4). Based on our observation that many of these biotinylations occur in solvent-occluded regions, we reasoned that biotin

'painting' is likely to happen during the intermediate steps of ribosome assembly during which otherwise hidden lysine and tyrosine residues would be exposed or during reversible dissociation of individual ribosomal proteins that can interact with the small (18S) or large (28S) ribosomal RNA hubs (Supplementary Fig. 4). Alternatively, it is also possible that biotinylation occurs on nascent chains of ribosomal proteins as they emerging from translating ribosomes. The fact that we observe a statistical enrichment of sites of biotinylation within IDRs in ribosomal proteins suggests that promiscuous biotinylation of nascent chains outside of IDRs is infrequent (Supplementary Data file 1). Several endogenous PTMs of ribosomal subunits, including acetylation, phosphorylation and methylation have recently been identified and further support the notion that the large surface of ribosomal proteins (Fig. 4 and Supplementary Fig. 4) can be used for efficient post-translational modifications (Supplementary Data file 1)[61]. Briefly, we observed extensive biotinylation in ribosomes, mostly in their IDRs, consistent with expected high flexibility of ribosomes and previously observed dense modification with multiple types of other PTMs.

We envisage many possible benefits from our novel biotin painting assay concept for exploring in vivo structure-functional questions; Firstly, to complement very detailed kinetic in vitro studies that can resolve conformational dynamics at high spatial and temporal resolution using hydrogen deuterium exchange (HDX). Biotin painting could enable complementary in vivo comparisons of the same target proteins and thereby increase the scope of HDX or related protein surface accessibility-based structural proteomics techniques[62–64]. Secondly, to acquire dynamic snapshots of biological pathways and determine by which mechanism these rewire biomolecular interaction networks and modulate subcellular conformations of proteins as recent technological advances both in biotinylation enzymes and multiplexed mass analysis will accelerate sampling of more biological timepoints[65–67]. Thirdly, to study dynamic in vivo drug effects. Many new drug candidates are failing in the later stages of development due to our incomplete understanding of cellular biology. If we can re-purpose BioID or other biotinylation methods for elucidating subcellular protein interactions, we might achieve earlier insights into drug (in)efficiency in relevant biological contexts.

Our analyses were enabled by the precise identification of sites of biotinylation using peptide-level enrichment which are not typically captured in more widely used protein-level experiments. An obvious limitation of peptide-level enrichment is that non-biotinylated peptides cannot contribute to the mass spectrometric signal, which can mean that more biological input material may be required in some cases. While peptide-level enrichment increases the specificity and analytical efficiency for detecting biotinylated peptides[39–42], it comes at the expense of not being able to detect proteins that lack lysines or detectable peptides with one missed cleavage (due to a modified lysine). Sequence coverage might be improved by including additional proteases in future biotin-based proximity experiments[68].

A key assumption in classical proximity proteomics studies is that biotinylation is enhanced near the biotin-activating enzyme. Our study shows that unfolded regions can be more readily biotinylated compared to folded regions. This could mean that proteins that in reality never change their cellular distribution can be perceived as spatially further away or closer to a BirA-fusion due to condition-dependent local folding or unfolding, respectively. We do not currently have definitive answers on how to unambiguously dissect condition-dependent local (un)folding from subcellular redistributions. It appears worthwhile to envisage the possibility that transient changes in protein folding can be important modulators of cellular dynamics that should be

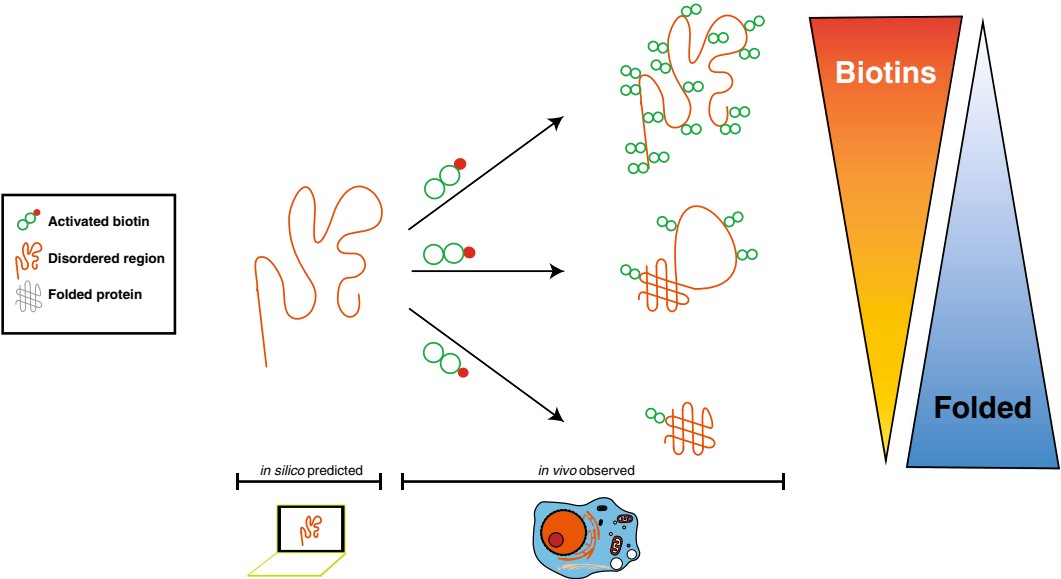

**Fig. 5 Biotin tagging favours unfolded protein regions in cells.** Biotinylation are more likely in predicted IDRs suggesting that they are more (at least transiently) accessible for biochemical modifications compared to folded proteins. Fully folded proteins can also be modified but show lower fractions of modified residues compared to IDRs. This positive correlation of biotinylation density and IDRs, i.e. biotin painting IDRs can be used to re-purpose biotinylation-based proximity proteomics studies to monitor protein plasticity in vivo.

**Table 1 Sources of data used in this study, Supplementary Information (SI).**

| Study | Ref | Target | Chemistry | Data Source | Concentration of activated biotin [μM] |
|---|---|---|---|---|---|
| BioSITe | 39 | Tyrosine | APEX2 | SI file 2, Supplementary Table 8 | <50 |
| Ab-APEX | 40 | Tyrosine | APEX2 | SI Table 6 | <500 |
| DiDBiT | 41 | Lysine | NHS-Biotin | SI file 2, Table S27 | 1000 |
| SpotBioID | 42 | Lysine | BirA | SI file 2, sheets 2–5 | <50 |

more broadly factored into experimental designs of proteomics studies (Fig. 5).

In summary, we have shown using several orthogonal analyses that in vivo biotinylation occurs at a greater rate within predicted IDRs and that highly disordered proteins are more likely to be biotinylated than those that are mostly folded. Furthermore, this trend of increased biotinylation in IDRs is not dependent on the algorithm we use to predict IDRs. However, the greater sensitivity of VSL2b enables the establishment of the trend also in short regions of local disorder and leads to a greater IDR fraction and more biotinylations assigned to IDRs. Finally, we have consistently observed that the SpotBioID study has more proteins that are highly disordered than the other three studies thereby validating previous predictions of large fractions of IDRs in nuclear proteins in vivo[54].

We envisage the possibility to repurpose cellular biotin tagging experiments to complement existing structural biology tools such as HDX-MS and provide fundamentally new insights into the cellular context of protein dynamics and interactions.

## Methods

**Source data description**. Four independent in vivo biotinylation studies have been used for our exploration of structural specificity of biotinylation sites. Their details are provided in Table 1 and they can be accessed as input files on the Github repository https://github.com/ComputationalProteomicsUnit/biotinIDR.

**Experimental input data to develop IDR predictions**. Some 60 published disorder prediction algorithms feature balanced accuracies of around 70–80%; some being designed and validated to predict short IDRs (<30 residues) and others being

better at determining long or both long and short IDRs[69]. The majority of these predictors are trained on a limited set of in vitro structural data, mainly X-ray crystallography data, Nuclear Magnetic Resonance (NMR) mobility data in the DisProt database (http://www.disprot.org/)[70].

**D2P2 resource description and selection of predictions**. A subset of more frequently used prediction algorithms has pre-computed predictions in the web resource D2P2 (http://d2p2.pro/[56]). D2P2 also offers the option to select a consensus call for IDRs in a given protein that is predicted by most of the nine different compound predictors. Of the nine callers included in D2P2, we focused our interest initially on the two most orthogonal callers: VSL2b which has high sensitivity for calling IDRs in both short and long regions of IDR[71] and IUPRed-L which has been trained to predict long disorders with high confidence[57]. As additional comparisons, we also predicted IDRs using a combination of VSL2b and IUPRed-L where an IDR was accepted if called by one or both predictors and consensus of (at least 75% of) nine predictors included in D2P2 (Supplementary Fig. 3).

**Assigning disorder predictions**. For all versions of IDR calling, we did not set any restrictions on the length of IDR. This means that an IDR can be called on a residue of length 1. While this might yield a lot of false positives, we wanted to ensure sensitivity rather than specificity of IDR calling. Having tested the four different versions of IDR calling with D2P2, we realised consistent trends between all predictions approaches while higher local sensitivity of VSL2b enabled more insights on local disorder. We, therefore, performed more detailed biotin site and PTM analyses using VSL2b. The IDR assignment uses an Application Programming Interface (API) to the D2P2 website and code to use this API was kindly provided by Dr. Tom Smith. The scripts d2p2.py and protinfo.py are necessary for the final analysis and can be accessed through the github repository https://github.com/TomSmithCGAT/CamProt/tree/master/camprot/proteomics. The python script for the final IDR analysis and output is called Get_IDRs-DM-v2.py and can be accessed via the repository https://github.com/ComputationalProteomicsUnit/biotinIDR.

**Protein sequence modification or proteoform images**. Images summarising the location of IDRs and PTMs were produced using Protter (http://wlab.ethz.ch/protter/) and protein structure images were generated using Pymol. For Protter images, scripts were written to generate an appropriate URL and then batch download it from the server. These scripts printUrl.py and runUrl.sh are also available via the Github repository https://github.com/ComputationalProteomicsUnit/biotinIDR (Supplementary Data 1, Protter-List).

**Statistical tests used**. To compare expected rates of biotin/PTMs and observed counts, we used a standard binomial test in R (binom.test). For estimating the background rate of biotins, we counted all the lysines (K; BirA based studies) or tyrosines (Y; APX based studies) in the protein sequence and within predicted IDR regions. For estimating the background rate of PTMs, we counted all the lysines (K; ubiquitination, acetylation, sumoylation) or serines, threonines and tyrosines (S, T, Y; phosphorylation) in the protein sequence and within predicted IDR regions. We defined the probability of success as the number of residues in IDRs/Total number of residues, a success as a biotin or PTM within an IDR and number of trials is the number of Biotins or PTMs observed in that study.

We applied a similar test to the one above look for enrichment of biotins specifically in ribosomal proteins, separated by cytosolic ribosomal proteins and mitochondrial ribosomal proteins using their protein names.

To look for differences in PTMs and biotins in the three protein groups—Folded, Partially Folded and Unfolded, we used pairwise *t*-tests or ANOVA followed by a post-hoc correction of family-wise error rates using a Tukey's Honestly Significant Differences test. The former yields a *p*-value while the latter yields a confidence interval for the effect size as well as a *p*-value. To compare number of biotins and IDRs, we used a standard Pearson's correlation test. To compare mean PTMs between the HEK293 biotinome and HEK293 proteome we used a standard *t*-test for means.

To perform a GO enrichment analysis, we used the package goseq[72] which is based on a Hypergeometric test with a Wallenius' correction which accounts for any biases in the data such as gene length, protein expression etc. In our study, we used protein expression from Geiger et al.[73], as the bias factor prior to calculating GO enrichment.

**Biomolecular structure visualisation**. The Cryo-EM structure of the human 80S ribosome (PDB ID 4v6x[74]) was visualised using ChimeraX[75]. SERBP1 and its biotinylated sites were highlighted using the sel function in its command line interface. RNA was coloured purple and protein subunits (except SERBP1) blue. All ribosomal macromolecules were visualised in surface representation. The FK506- and rapamycin-binding protein 3 (FKBP3) NMR structure (PDB ID 2mph) was visualised in cartoon model of the first low energy model; surfaces were kept 90% transparent except around biotinylation sites that were highlighted in yellow. Bacterial 70S ribosomes were visualised using white for proteins and pale turquois for nucleic acids and red for sites of biotinylation.

**Bacterial 70S ribosome purification**. Strain JE28[76] was grown in 2 x YT media with 50 μg/ml kanamycin in baffled flasks at 37 °C to OD$_{600 nm}$ of 1.0 and harvested by centrifugation and resuspended in 'High Mg$^{2+}$ buffer A' (50 mM HEPES pH 7.0; 20 mM MgCl$_2$; 50 mM NaCl; 50 mM KCl; 100 mM NH$_4$Cl; 5% glycerol; 1 tablet per 250 mL of cOmplete protease inhibitor cocktail tablet (Roche). Cells were flash frozen with liquid nitrogen and stored at −80 °C. A thawed cell suspension was lysed with emulsiflex at 15,000 psi and clarified by centrifugation at 32,000 × *g* for 30 min at 5 °C.

For 70S, the lysate was applied to HiTrap chelating resin with bound Ni 2+ in high Mg$^{2+}$ buffer and eluted with gradient 0–100% 'High Mg$^{2+}$ buffer B' (50 mM HEPES pH 7.0; 20 mM MgCl$_2$; 50 mM NaCl; 50 mM KCl; 100 mM NH$_4$Cl; 5% glycerol; 1 tablet per 250 mL of cOmplete protease inhibitor cocktail; 0.5 M imidazole, adjust pH to 7.0).

The 70S particles were concentrated and applied to a preparative S200 column (S200 buffer: 50 mM HEPES pH 7.0; 20 mM MgCl$_2$; 75 mM KCl; 75 mM NaCl; 5% glycerol). While the matrix was not optimal for removing higher MW contaminants, it was effective to remove contaminants smaller than the 30S, 50S and 70S.

**Biotin painting the bacterial 70S ribosome**. To test fast and slow biotinylation, 70S ribosomes were incubated with 1 mM NHS-biotin in triplicates for 10 s, 120 s, 300 s and 1 h. The 1 h timepoint was used as a carrier reference and excluded from statistical analyses. The incubation took place at 37 °C in a buffer containing 20 mM MgCl2 (and 50 mM HEPES pH 7.0; 75 mM KCl; 75 mM NaCl; 5% glycerol) and then quenched using 60 mM hydroxylamine for 15 min. Biotinylated ribosomes were heated to 95 °C for 10 min in 6 M guanidine hydrochloride, 5 mM TCEP, 10 mM Chloracetamide to denature proteins and alkylate cysteine residues. Contaminants and biotin were removed using the modified SP3 protocol, briefly by precipitation using 20:1 (v/v) ethanol to peptide-bead mix and three subsequent washes using 50 bead volume equivalents of 80% ethanol washing off the magnetic beads as previously reported[77]. Samples were digested overnight at 37 °C with 1.2 μg modified trypsin-LysC (Promega). TMT labelling was performed directly on beads using 60 μg per sample and channel using 10-plex (Thermo Fisher Scientific)

using a sufficient molar excess of TMT label in HEPES buffer as previously described[78].

Mass spectra were acquired in positive ion mode applying data acquisition using synchronous precursor selection MS3 (SPS-MS3) acquisition mode as in Queiroz et al.[79,80]. Samples were analyzed in an Orbitrap Fusion Lumos (Thermo Fisher Scientific), coupled to a 50 cm long PepMap nanoLC column (on a Dionex Ultimate 3000 UHPLC). All samples were analyzed in a 120-min gradient from 9–45% buffer B (containing 80% acetonitrile) and SPS-MS3.

**MS spectra processing and peptide and protein identification**. Raw data were processed using Proteome Discoverer v2.3 (Thermo Fisher Scientific). The raw files were submitted to a database search using Proteome Discoverer with Mascot and SequestHF algorithms using the *E. coli* database downloaded in early 2017, UniProt/TrEMBL. Common contaminant proteins (several types of human keratins, BSA and porcine trypsin) were added to the database, and all contaminant proteins identified were removed from the result lists before further analysis. The spectra identification was performed with the following parameters: MS accuracy, 10 p.p.m.; MS/MS accuracy of 0.05 Da for spectra acquired in Orbitrap analyzer and 0.5 Da for spectra acquired in Ion Trap analyzer; up to two missed cleavage sites allowed; carbamidomethylation of cysteine (as well as TMT6plex tagging of lysine and peptide N terminus for TMT labelled samples) as a fixed modification; and oxidation of methionine and deamidated asparagine and glutamine as variable modifications. Percolator node was used for false discovery rate estimation and only rank 1 peptide identifications of high confidence (FDR < 1%) were accepted. TMT reporter values were assessed through Proteome Discoverer v2.3 using the Most Confident Centroid method for peak integration and integration tolerance of 20 p.p.m. Reporter ion intensities were adjusted to correct for the isotopic impurities of the different TMT reagents (according to the manufacturer specifications for the respective batch number).

**TMT data analysis**. The data obtained from Proteome discoverer was abundance data at the peptide level. We filtered the data to remove missing values and any non-biotinylationed peptides as well as peptides that were non-ribosomal. The data were not median-normalised before the statistical analysis because we did not want to introduce normalization artifacts by (artificially) equalizing actually very different fractions of biotinylated peptides within the total peptide pools in TMT channels corresponding to 10 s and 300 s in our 10-plex TMT set (note: such a normalization would make sense after specific enrichment for biotinylated peptides).

We used the R[81] package limma[82] to test the significant increase of labelling between triplicate measurements at 10 s and triplicate measurements at either 120 s or at 300 s. TMT reporter ion intensities of biotin-peptides at 10 s were used as reference and any significant increases (using adjusted *p*-values smaller than 0.05) were selected as 'late' biotinylation events while those that did not statistically increase further after the 10 s timepoint were designated as early biotinylation events and used for mapping of biotinylated sites within the 70S cryoEM structure.

**Reporting summary**. Further information on research design is available in the Nature Research Reporting Summary linked to this article.

## Data availability

All TMT-multiplexed mass spectrometry data have been deposited to the ProteomeXchange Consortium via the PRIDE partner repository[83] with the dataset identifier PXD016422. Details for the structural mapping of sites of biotinylation are described in the Supplementary Dataset 3 and Supplementary Figs. 6 and 7.

## Code availability

Code to process biotinylation datasets and reproduce the analysis has been deposited in the Github repository https://github.com/ComputationalProteomicsUnit/biotinIDR. Please request access to the code by emailing the corresponding authors as it will be made public following journal acceptance.

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

## Acknowledgements

We wish to thank Prof. Keith Dunker and Dr. Sudhakaran Prabakaran for inspiring discussions, advice and comments on earlier versions of this paper. We thank Dewi Eburne for help with exploratory experiments that triggered our interest in structural features of biotinomes, Prof. Ben Luisi for the kind gift of highly purified 70S ribosomes, Dr. Tom Smith for kindly sharing his code for IDR parsing and GO term extension. We thank Dr. Matthew Young for contributing his ideas on complementary data analysis strategies. D.P.M. is supported by the BBSRC grant BB/N010493/1 and a gold level TMT grant (Thermo Fisher, 2018) for studying in vivo biotinylation dynamics. We would like to thank Mike Deery for excellent technical support with the mass spectrometry for this project.

## Author contributions

D.P.M., M.R. and K.S.L. conceived the design of the study. D.P.M. and M.R. analysed the data. M.R. mapped and quantified correlations of IDRs, PTMs and sites of biotinylations. D.P.M. inspected sites of biotinylation mapped on experimentally solved high-resolution structures. K.S.L provided insightful feedback along all stages of the project. D.P.M., M.R. and K.S.L. wrote the paper. D.P.M and M.R contributed equally to the study.

## Competing interests

The authors declare no competing interests.
