## [Peer Review File · Communications Biology]

Reviewers' comments:

Reviewer #1 (Remarks to the Author):

In "Biotin proximity tagging favours unfolded proteins", Minde et al. summarized the statistical analysis on the biotin sites identified in four independent studies, and concluded the significant preferences of biotin sites in the intrinsically disordered regions. This is an interesting study, given that the IDRs are critical for many health problems, because the modification in the IDRs is believed to affect their functions.

Here are some major issues:

1. More evidence is needed to support the conclusion in this paper. In particular, in the four datasets they used, the DiDBiT does not agree with the conclusion (Figure 3B), showing smaller observed percentage than the expected one. Yet, this dataset has the largest number of biotin sites (11330), more than all others combined (Figure S2A). This desires more explanations.

2. The other issue is regarding the IDR predictions, although many algorithms claim to be 70% or better accuracy, it is still a challenging problem since they are not sufficient validation data for those algorithms at the proteome level. Drawing conclusions from a few prediction algorithms can be dangerous. The authors should consider to use some consensus predicted results (being more conservative), and gradually relax the constraints by using fewer algorithms (i.e., the bigger set of consensus) to estimate the prediction error rate.

3. Using the p-values computed from the binomial test is not well justified to indicate the statistical significance.

A few minor issues for consideration:

1. The authors computed the expected rates, as described "we calculated the number of lysine residues (K; for SpotBioID and DiDBiT) or tyrosine residues (Y; Ab-APEX and BioSITE)", and in the Figure S2A, only lysine information is used. Furthermore, it is unclear whether the accessible areas (probabilities) in the disordered or folded regions are treated differently. For example, lysine in the IDRs may be considered as 100%, while in the folded region is 50% accessible.
2. Figure 3C shows the percentiles of three forms of proteins. How well do the results from these two algorithms overlap? The uncertainty in the prediction may affect the conclusion.
3. The example of ribosome does not support that IDRs are favorable for biotin sites. At least from the Figure 4, this message is not well conveyed. Using a ribbon representation for subunits might be better, surfaces can be used in their transparent form.
4. Figure2 legend: Fig. 2 Illustrative examples for in vivo surface biotinylation is four independent studies. -> "in four independent studies"
5. What's the point of repeating 2A (i-iv)? A plot similar to multiple sequence alignment can show better differences. Legneds should removes the cross sign in each symbol.

Reviewer #2 (Remarks to the Author):

Summary:

Proximity based labeling is emerging as a powerful means to assess the spatial interactome in vivo. In this manuscript, the authors attempt to determine whether different labeling techniques such as BioID or APEX selectively label, and therefore identify, proteins with Intrinsically Disordered Regions (IDRs). By comparing four different proteome wide approaches with these various techniques, using in silico

analyses only, they are able to determine both at the specific protein level and proteome wide, that labeling occurs more commonly at IDR's and that this is a feature present across all methodologies. These findings are exciting and important for biologists using these techniques to better understand the strengths and limitations of proximity based labeling. However, this manuscript is limited by insufficient data for each methodology to be able to support their conclusions (single study each, different localizations for each protein), and a lack of discussion of the unique attributes for each approach.

Major points:

1. Mass spec data. Given that each of the four datasets are unique in terms of labeling strategy and bait type it would be more powerful to compare data with the same technique against each other. Are there additional published datasets that could be used to have an independent replicate for at least one of the techniques? Can specific peptides listed in BioID and APEX papers be analyzed for localization within IDR? How generalizable are the findings from these 4 studies?
2. Example protein labeling. In figure 2 A and B, both Emerin and SERBP1 have shared labeling sites across techniques and unique sites for each technique. Can the authors provide information about the neighboring amino acids and structure to better delineate why labeling may or may not have occurred at these locations? Can a nuclear or cytoplasmic only protein (from the shared 29) also be included to further investigate potential biases in labeling?

Minor points:

1. Figure 3B. Can the authors speculate why DiDBiT did not show a preference for IDR as compared to the other techniques? The fact that DiDBiT does not show a preference is concerning since this method has no bait-based bias and should provide a more uniform view of biotin accessibility.
2. How are metabolic proteins that bind biotin dealt with in this paper? Are these found among the shared 29 proteins across all 4 studies?
3. Figure 3C. Can the authors provide a column with predictions of IDR for all proteins identified in an input sample for mitochondria, cytoplasm, nucleus and whole cell extracts? This would provide a clearer rationale for stating an enrichment of unfolded proteins using these techniques- currently it is not clear what enrichment the authors are referring to.

David B. Beck MD, PhD
Postdoctoral Fellow
National Institutes of Health

Reviewer #3 (Remarks to the Author):

In the research paper entitled 'Biotin proximity tagging favours unfolded proteins', Minde et al explored the association between positional biotinylation by proximity labelling and intrinsically disordered regions (IDRs) in proteins. Particularly, the authors correlated the biotinylation positions of >20000 proteins that were found post-translationally modified in four independent orthogonal large-scale biotin proximity tagging studies and IDRs predicted by a series of algorithms implemented in a web resource. The authors claimed to have observed increased biotinylation density in predicted IDRs and conclude that biotin tagging can be used to predict the stability of IDRs in proteins in vivo.

The study shows novelty to some extent, as the same authors state in the manuscript (lines 59-66), it was already reported in the literature that increased post-translational modifications (PTMs) of different kind correlate with IDRs in proteins. Considering the upraising interests on intrinsically

disordered proteins (IDPs), the study might be of interest, at least to the niche fields of structural and molecular biology.

Generally, the manuscript is clear and neat figures are reported. However, I think re-editing will be required to obtain more robust significance and better flow.

Introduction

Line 23: the authors cite alternative splicing as a way to expand the cells proteome. Even though this is unequivocally true, I think it is not relevant to cite mRNA processing in a paper that is focused on post-translational modifications of proteins.

Lines – 48-58: the authors state that variations of pH, salt concentrations and PTMs can have effects on the conformational ensembles of IDPs. To support this statement, two examples about the effect of salt concentration and temperature on proteins solubility and stability are reported. Considering the interest of the research reported in the paper, I believe it would be more relevant to report examples from the previous literature on the effects of PTMs rather than ionic strength/temperature on ensembles of IDPs (examples are available in Bah et al J Biol Chem. 2016 Mar 25; 291(13): 6696–6705).

Methods

I believe the heading of this section for Communications Biology is 'Methods' rather than 'Material and methods'.

Lines 412-438: this section, which describes the methods used to assign disordered regions in protein, is quite lengthy. The sentence between lines 413-417 could be probably summarised, while the section between lines 427-435 describes experimental results rather than methods.

Results

The 'Results' starts with a section entitled 'Concept of the study', which summarises the main aim of the research (identifying a correlation between increased biotinylation and in vivo stability of IDRs of disease-linked proteins). Even though I appreciated the recap to guide towards the results analysis, I found this section too lengthy. I believe part of this section should be implemented in the final paragraph of the introduction (lines 85-92), while the 'Results' should start with a much shorter statement on the aims of the study.

The following section of the 'Results', entitled 'Brief introduction to selected proximity proteomics studies', reports a detailed description of the four independent orthogonal large-scale biotin proximity tagging studies used in this research. This description is absolutely necessary to allow the reader to understand how the four studies have been chosen and what kind of results included. However, I believe it is out of context in the 'Results' (it is not original results) and it should be moved as it is in 'Methods'. In the 'Results', only a very short description of the four studies should be provided (to allow a fluid reading), referencing Fig 1B.

Next, a section entitled 'Orthogonality of tyrosine and lysine as molecular targets of proximity proteomics' describes the different solvent accessible surface areas of tyrosine and lysine in proteins undergoing PTM. Again, such section does not report any kind of experimental result and it rather describes information previously described in the literature. The section should be therefore moved to either 'Introduction' or 'Methods'.

Line 245: the acronym for FKBP3 is missing.

Conclusion

This section should be part of the 'Discussion', as per Communications Biology research paper structure.

Fig2

The panel 2A's legend is reported in 2B.

The work is overall convincing; however, I have few major concerns.

From what reported in Fig2Av and 2Bii, there is very low correspondence of the biotinylation positions identified for the two proteins Emerin and SERBP1 by the 4 different large-scale biotin proximity tagging studies. This would suggest that the biotinylation is (at least partially) a non-specific phenomenon. Can the authors comment on this? Is there any example among the mentioned 29 orthogonally selected proteins where biotinylation is highly specific among the four studies and whose position highly correlate with IDRs? If so, this should be included in the list of examples reported in the 'Results' section called 'Illustrative examples of proteins that are biotinylated across all four independent studies', to support the initial hypothesis. If not, I believe this should be discussed. Is there any correlation between the proteins whose IDRs are biotinylated and post-translationally modified by different tags (sumo, phosphates, ubiquitin, acetylation)? If so, statistics should be provided to reinforce the study and corroborate that also biotinylation can be used to predict IDRs. Otherwise, discussion is anyway required.

The authors report the case of 80S ribosomal complex to support their hypothesis that biotinylation is increased in IDRs of proteins. They state that «virtually all biotinylated subunits are non-globular and contain many biotinylation». I am afraid that the statement is not strong enough to support the hypothesis and a thorough analysis should be presented. To really corroborate the hypothesis, the authors must provide data indicating significant correlation between biotinylation observed in the subunits of the 80S ribosomal complex and IDRs of such proteins (predicted or observed). Also, as the authors state, «Many of these (proteins) are inaccessible to water or larger molecules such as biotin in the fully assembled 80S ribosomal complex as they are contacting ribosomal RNA». This leads to the statement in the 'Discussion' that the biotinylation happens before or during assembling of the 80S ribosomal complex. Can the authors expand on the biological relevance/significance of this? Has previously ever reported in the literature that post-translational modification of the subunits is necessary for the 80S ribosomal complex?

Reviewers' comments:

Reviewer #1 (Remarks to the Author):

In "Biotin proximity tagging favours unfolded proteins", Minde et al. summarized the statistical analysis on the biotin sites identified in four independent studies, and concluded the significant preferences of biotin sites in the intrinsically disordered regions. This is an interesting study, given that the IDRs are critical for many health problems, because the modification in the IDRs is believed to affect their functions.

Here are some major issues:

1. More evidence is needed to support the conclusion in this paper. In particular, in the four datasets they used, the DiDBiT does not agree with the conclusion (Figure 3B), showing smaller observed percentage than the expected one. Yet, this dataset has the largest number of biotin sites (11330), more than all others combined (Figure S2A). This desires more explanations.

We agree that the DiDBiT study data appear as an 'outlier' within our analysis and agree that additional explanations are necessary to explain the likely reasons behind this observation and we have now discussed these in the revised manuscript (p. 7, lines 294-298) One explanation is the fact that the DiDBiT study used extended labelling times that may have resulted in a broader range of biotinylation events. To support this explanation, we have now added additional experimental dataset that we have collected to address the specifics of the labelling patterns in NHS-biotin for extended periods of time. In technical replicates of time-courses spanning several orders of magnitude (10 seconds to 3600 seconds), we observed that short incubation times result in the highest preference for disordered regions (30% of biotinylation events at 10 seconds of labelling), whereas longer time points resulted in promiscuous labelling that penetrated deep into structured domains of the 70S bacterial ribosome complex. We conclude that unlike in more limited biotinylation events such as brief pulses typical of APEX2 based experiments or slow BirA biotinylation in BioID experiments, the extensive (one hour long) labelling using 1 mM NHS-biotin in the DiDBiT experiment leads to saturation effects that obscure the intrinsic preference of biotinylation for disordered regions.

2. The other issue is regarding the IDR predictions, although many algorithms claim to be 70% or better accuracy, it is still a challenging problem since they are not sufficient validation data for those algorithms at the proteome level. Drawing conclusions from a few prediction algorithms can be dangerous. The authors should consider to use some consensus predicted results (being more conservative), and gradually relax the constraints by using fewer algorithms (i.e., the bigger set of consensus) to estimate the prediction error rate.

We completely agree that all IDR prediction approaches are not 100% accurate and that proteome level validation is still challenging. As suggested, we have now explored consensus predictions and individual prediction tools (see. S2A) and have added a new Figure (Fig. S3). Some of the variability of disorder predictions is due to different definitions of disorder across different IDR prediction algorithms. Some algorithms attempt to identify globular folded domains or their absence (IUPred). Other methods, including VSL2b, are more broadly looking at non-rigid structures both globally and locally, for instance in short flexible protein segments as evident in missing local density in x-ray experiments or high variability in NMR data or CryoEM structures. The D2P2 'consensus' is better at identifying regions in proteins that lack folded domains, while it typically misses locally more dynamic segments within otherwise folded domains. Our preference for IUPred and VSL2b can be justified because:

- (i) 'consensus' is insensitive for local IDRs but local disorder can be functionally important
- (ii) IUPred is similar in detecting folded segments as a 'consensus' tool as shown with additional tests (new Supp Fig. S3, Table S2A)
- (iii) IUPred (unlike 'consensus') has a traceable definition of disorder and is therefore easier to understand for the general reader than the 'consensus' that is a mix of all prediction.

Overall, we feel that currently there is no 'gold-standard' measure for disorder prediction. In other words, it is not possible to precisely quantify the accuracy as the 'true positives' and the 'true negatives' of cellular IDRs are yet to be clearly defined. Existing tools cover different aspects of the phenomenon. Continued community-wide experimental and computational efforts will hopefully yield novel condition-specific tools with increased proteome-wide IDR-prediction accuracy for living cells. This is however clearly outside the scope of this study.

3. Using the p-values computed from the binomial test is not well justified to indicate the statistical significance.

We set out to test whether we see more biotins/PTMs within a region of IDR than expected by chance. This meant that for every biotin in each of the studies, we wanted to know whether it was within an IDR (success) or

outside an IDR (failure). Thus, the definition of this problem very much yielded itself to a binomial test, particularly as we were able to calculate the probability of success based on the number of lysine and tyrosine residues within the region of IDR. Once we had a base probability, we tested to see whether what we observed was greater than what we expected (one-sided binomial test). By all means, we are open to suggestions of using a different named test but we cannot see an obvious error in using a binomial test for the hypothesis we were testing.

A few minor issues for consideration:

1. The authors computed the expected rates, as described “we calculated the number of lysine residues (K; for SpotBioID and DiDBIT) or tyrosine residues (Y; Ab-APEX and BioSITE)”, and in the Figure S2A, only lysine information is used.

This is correct. We thank the diligent reviewer for pointing out this inconsistency and have now corrected this typo in S2A, which should read ‘target amino acid’ (we have double-checked the code used for analysis and can confirm that the analysis is correct for both lysine-targeting and tyrosine-targeting datasets).

Furthermore, it is unclear whether the accessible areas (probabilities) in the disordered or folded regions are treated differently. For example, lysine in the IDRs may be considered as 100%, while in the folded region is 50% accessible.

We have NOT imposed a different weighting factor for accessibility in IDRs as the cellular accessibility of IDRs is an unknown except for one protein (α -synuclein, <https://doi.org/10.1038/nature16531>). We have made this more explicit (page 26, lines 882-884).

2. Figure 3C shows the percentiles of three forms of proteins. How well do the results from these two algorithms overlap? The uncertainty in the prediction may affect the conclusion.

As stated above, we have also used the ‘consensus predictor’, which shows that at least for SpotBioID and BioSITE, all predictors agree and that most biotinylation maps to partially or mostly disordered regions for these studies. We have added this on page 10 (lines 324-325) and added a new supplementary figure (Fig S3) to support this statement.

3. The example of ribosome does not support that IDRs are favorable for biotin sites. At least from the Figure 4, this message is not well conveyed. Using a ribbon representation for subunits might be better, surfaces can be used in their transparent form.

As stated earlier (in cited ref 62) the 80S ribosome contains rather untypically shaped non-globular proteins. Our data support the findings of this publication. If indeed the assumptions about the non-globular dynamic disorder of the ribosomal subunits were incorrect, then one would expect to see no biotinylation in RNA-protein interfaces. This clearly is not the case and to further reinforce our findings we have added another figure to make this clearer (Supp Fig. S4).

4. Figure2 legend: Fig. 2 Illustrative examples for in vivo surface biotinylation is four independent studies. -> “in four independent studies”

This has been corrected

5. What’s the point of repeating 2A (i-iv)? A plot similar to multiple sequence alignment can show better differences. Legends should remove the cross sign in each symbol.

This has now been corrected in a new version of Figure 2 without cross signs. We have used the multi-coloured Protter representations because this makes it immediately clear to the general reader that Emerin is a membrane protein and that its membrane-extrinsic regions can be decorated with multiple PTMs in different subcellular niches. A(v) and B(ii) also provide the requested MSA-like representations. While this might be considered redundant by highly specialised readers who know very well that Emerin is an intrinsic membrane protein, we feel that this alternative representation adds valuable information for the general reader.

Reviewer #2 (Remarks to the Author):

Summary:

Proximity based labeling is emerging as a powerful means to assess the spatial interactome in vivo. In this manuscript, the authors attempt to determine whether different labeling techniques such as BioID or APEX selectively label, and therefore identify, proteins with Intrinsically Disordered Regions (IDRs). By comparing four different proteome wide approaches with these various techniques, using in silico analyses only, they are able to determine both at the specific

protein level and proteome wide, that labeling occurs more commonly at IDR's and that this is a feature present across all methodologies. These findings are exciting and important for biologists using these techniques to better understand the strengths and limitations of proximity based labeling. However, this manuscript is limited by insufficient data for each methodology to be able to support their conclusions (single study each, different localizations for each protein), and a lack of discussion of the unique attributes for each approach.

Major points:

1. Mass spec data. Given that each of the four datasets are unique in terms of labeling strategy and bait type it would be more powerful to compare data with the same technique against each other. Are there additional published datasets that could be used to have an independent replicate for at least one of the techniques? Can specific peptides listed in BioID and APEX papers be analyzed for localization within IDR? How generalizable are the findings from these 4 studies?

We thank the reviewer for pointing out the importance of replicates. We completely agree with this but were limited by available studies for which there are no obvious replicate datasets.

Nearly all published BioID and APEX studies that report large numbers of proteins, lack specific biotinylation site information perhaps due to technical challenges for specific and complete elution of biotin-peptides. We therefore opted to acquire our own additional biotinylation replicate data

(Supp. Fig. S5, S6)

2. Example protein labeling. In figure 2 A and B, both Emerin and SERBP1 have shared labeling sites across techniques and unique sites for each technique. Can the authors provide information about the neighboring amino acids and structure to better delineate why labeling may or may not have occurred at these locations? Can a nuclear or cytoplasmic only protein (from the shared 29) also be included to further investigate potential biases in labeling?

We have determined if there is any consensus subcellular niche between any of the shared proteins (see new supplementary table 2). Only endogenously biotinylated Pyruvate Carboxylase had a direct consensus location by all analysed sources of subcellular location information (HPA/Uniprot/LOPIT-DC). We have added this finding as Table S2.2 and cited the corresponding papers on page 5 (lines 191-197). This analysis reveals that indeed these proteins have been found in more than one subcellular location, except for endogenously biotinylated mitochondrial Pyruvate Carboxylase.

We have also analysed the neighbouring amino acids distributions around all detected sites of biotinylation but did not include this analysis in the manuscript as there is no relevant enrichment. The only exception is tryptic sites in the vicinity of biotinylation events, which is consistent with the expected technical bias due to higher efficiency of fragmentation of short peptides and therefore higher identification rates compared to very long peptides. We feel that this that information is trivial and of little interest to the specialist reader and distracting for the general reader. The 29 shared proteins are too few to perform statistical tests on neighbouring amino acids.

Minor points:

1. Figure 3B. Can the authors speculate why DiDBiT did not show a preference for IDR as compared to the other techniques? The fact that DiDBiT does not show a preference is concerning since this method has no bait-based bias and should provide a more uniform view of biotin accessibility.

Please see the above comment to reviewer #1. We have added our own experimental data that highlight the outcome of extended labelling times of the extent of biotinylation as observed in the DiDBiT study. Indeed, we note that long incubation with biotin leads to promiscuous labelling also in structured regions of proteins.

2. How are metabolic proteins that bind biotin dealt with in this paper? Are these found among the shared 29 proteins across all 4 studies?

The metabolic enzyme, Pyruvate Carboxylase is a well characterised endogenously biotinylated protein and was detected in all four studies (see above). We did not further investigate this, as its enzyme catalysed biotinylation is well- documented, but added a remark in the discussion of the 29 shared proteins to highlight this fact for the less experienced reader (see above).

3. Figure 3C. Can the authors provide a column with predictions of IDR for all proteins identified in an input sample for mitochondria, cytoplasm, nucleus and whole cell extracts? This would provide a clearer rationale for stating an enrichment of unfolded proteins using these techniques- currently it is not clear what enrichment the authors are referring to.

We have provided all these data in the revised supplementary information excel sheet (new Table S1.4). We also performed additional statistical tests that show with high confidence that both PTMs and sites of biotinylation are enriched in four studies and the ribosomal subset (new Table S1.8).

David B. Beck MD, PhD
Postdoctoral Fellow
National Institutes of Health

We thank the reviewer for disclosing his identity.

Reviewer #3 (Remarks to the Author):

In the research paper entitled 'Biotin proximity tagging favours unfolded proteins', Minde et al explored the association between positional biotinylation by proximity labelling and intrinsically disordered regions (IDRs) in proteins. Particularly, the authors correlated the biotinylation positions of >20000 proteins that were found post-translationally modified in four independent orthogonal large-scale biotin proximity tagging studies and IDRs predicted by a series of algorithms implemented in a web resource. The authors claimed to have observed increased biotinylation density in predicted IDRs and conclude that biotin tagging can be used to predict the stability of IDRs in proteins in vivo.

The study shows novelty to some extent, as the same authors state in the manuscript (lines 59-66), it was already reported in the literature that increased post-translational modifications (PTMs) of different kind correlate with IDRs in proteins. Considering the upraising interests on intrinsically disordered proteins (IDPs), the study might be of interest, at least to the niche fields of structural and molecular biology.

Generally, the manuscript is clear and neat figures are reported. However, I think re-editing will be required to obtain more robust significance and better flow.

Introduction

Line 23: the authors cite alternative splicing as a way to expand the cells proteome. Even though this is unequivocally true, I think it is not relevant to cite mRNA processing in a paper that is focused on post-translational modifications of proteins.

We are grateful to reviewer #3 for pointing out that this section was not clear enough. PTM sites that are no longer present in alternatively spliced proteins can never be modified. 'PTMome' and 'spliceomes' are therefore intimately linked in biological outputs. We have attempted to make this point more explicit (by adding lines 28-30 on page 1)

Lines – 48-58: the authors state that variations of pH, salt concentrations and PTMs can have effects on the conformational ensembles of IDPs. To support this statement, two examples about the effect of salt concentration and temperature on proteins solubility and stability are reported. Considering the interest of the research reported in the paper, I believe it would be more relevant to report examples from the previous literature on the effects of PTMs rather than ionic strength/temperature on ensembles of IDPs (examples are available in Bah et al J Biol Chem. 2016 Mar 25; 291(13): 6696–6705).

We agree that an additional example of a PTM effect is informative and have included the suggested reference on page 2 (line 55). We have also expanded the related discussion of specific PTM examples such as phosphorylations, and their effect on protein stability as reported in the latest research on this topic (lines 62-66).

Methods

I believe the heading of this section for Communications Biology is 'Methods' rather than 'Material and methods'. Lines 412-438: this section, which describes the methods used to assign disordered regions in protein, is quite lengthy. The sentence between lines 413-417 could be probably summarised, while the section between lines 427-435 describes experimental results rather than methods.

We have corrected this.

While the information provided in lines 412-438 is arguably 'redundant' as there is prior literature on the subject, we feel that this section should be kept as guide to readers who are new to the IDR field and would benefit from knowing which data had been used in establishing these methods/algorithms. We have added additional subheadings ('Experimental input data to develop IDR predictions', 'D2P2 resource description' and 'IDR assignment strategy' on page 15 and 16) to facilitate the skipping of this section by more expert readers.

Results

The 'Results' starts with a section entitled 'Concept of the study', which summarises the main aim of the research (identifying a correlation between increased biotinylation and in vivo stability of IDRs of disease-linked proteins). Even though I appreciated the recap to guide towards the results analysis, I found this section too lengthy. I believe part of this section should be implemented in the final paragraph of the introduction (lines 85-92), while the 'Results' should start with a much shorter statement on the aims of the study.

We thank the reviewer for highlighting redundancies between the introduction and results section. We have eliminated repetitions with the introduction (lines 93-97).

We respectfully disagree, however, that a concept cannot be part of the results section as this paper is mainly about introducing our new concept of repurposing cellular biotinylation patterns for structural biology. Subsequent details on studies etc. support this and we believe are required for clarity. Putting these details in the introduction would be impractical and would require too much switching between sections. Readers with very detailed insights in structural biology and proximity methods can easily skip these well-marked sections as they have clear headings.

The following section of the 'Results', entitled 'Brief introduction to selected proximity proteomics studies', reports a detailed description of the four independent orthogonal large-scale biotin proximity tagging studies used in this research. This description is absolutely necessary to allow the reader to understand how the four studies have been chosen and what kind of results included. However, I believe it is out of context in the 'Results' (it is not original results) and it should be moved as it is in 'Methods'. In the 'Results', only a very short description of the four studies should be provided (to allow a fluid reading), referencing Fig 1B.

We thank the reviewer for the endorsement of our detailed description of approaches in the four studies we have re-analysed here. We respectfully disagree that moving Fig. 1B to other sections could improve fluid reading for most readers especially if they are not experts in proximity methods. Clearly it would not fit in the methods section as we did not perform this method ourselves and moving to the introduction would make this section too lengthy and disconnected in our opinion. We have however added a set of methods that relate to the new empirical quantitative data we collected to show the effect of time on the structural specificity of biotinylation.

We clearly mark the four studies as previously published and cite them in the figure legend. However, there are no comparable illustrations that contrast the biotinylation strategies as in our Figure 1 in the original papers. For example, in the SpotBioID study, the authors apparently overlooked that their bait must have been mostly located in the nucleus given that most biotinylation events took place on well documented nuclear proteins (see new Table S2). Furthermore, we want to explicitly contrast the subtle differences in approaches of these different studies which is only possible by directly comparing their key features in one figure.

Most importantly, the purpose of these illustrations is mainly to support the new concept of repurposing proximity proteomics studies for in vivo structural biology. We feel that the fact that these four studies report explicit sites of biotinylation and are highly specific to their target compartments according to our (new GO term and HPA/Lopit-DC comparison) analysis cannot be stressed too much. We have adjusted the section heading to make this clearer.

Next, a section entitled 'Orthogonality of tyrosine and lysine as molecular targets of proximity proteomics' describes the different solvent accessible surface areas of tyrosine and lysine in proteins undergoing PTM. Again, such section does not report any kind of experimental result and it rather describes information previously described in the literature. The section should be therefore moved to either 'Introduction' or 'Methods'.

We apologise if the purpose of this paragraph was not sufficiently clear. We have added a line to make it more explicit why this section is required to support the result of our new concept and related analysis (lines 173-175).

Line 245: the acronym for FKBP3 is missing.

We thank the reviewer for very diligent reading of the manuscript and have added an explanation of this acronym.

Conclusion

This section should be part of the 'Discussion', as per Communications Biology research paper structure.

We have corrected this.

Fig2

The panel 2A's legend is reported in 2B.

This has been corrected

The work is overall convincing; however, I have few major concerns.

From what reported in Fig2Av and 2Bii, there is very low correspondence of the biotinylation positions identified for the two proteins Emerin and SERBP1 by the 4 different large-scale biotin proximity tagging studies. This would suggest that the biotinylation is (at least partially) a non-specific phenomenon. Can the authors comment on this? Is there any example among the mentioned 29 orthogonally selected proteins where biotinylation is highly specific among the four studies and whose position highly correlate with IDRs? If so, this should be included in the list of examples reported in the 'Results' section called 'Illustrative examples of proteins that are biotinylated across all four independent studies', to support the initial hypothesis. If not, I believe this should be discussed.

We have added quantitative proteomics data that we have collected to address this question. High reproducibility of biotinylation sites even without enrichment suggests the low contribution of unspecific effects. The differences in biotinylation sites between studies is likely due to multi-localised proteins changing conformation and/or protein-protein interactions in different subcellular niches.

Is there any correlation between the proteins whose IDRs are biotinylated and post-translationally modified by different tags (sumo, phosphates, ubiquitin, acetylation)? If so, statistics should be provided to reinforce the study and corroborate that also biotinylation can be used to predict IDRs. Otherwise, discussion is anyway required.

Yes, we have performed an analysis to test the correlation between IDRs and the four post-translational marks that the reviewer mentions above. Our analyses show that there is a significant correlation between IDRs and Phosphorylation and Sumoylation but not with Ubiquitination or Acetylation. There were significantly more Phosphorylation and Sumoylation marks being identified in IDRs than expected (based on the rates of occurrence of S, T, Y and K) in the peptide sequences used in the analyses. We are happy to include the figures below as supplementary material if deemed necessary. Else, we have included a couple of lines in the discussion pertinent to this section and added the numbers to Table S1.8.

The authors report the case of 80S ribosomal complex to support their hypothesis that biotinylation is increased in IDRs of proteins. They state that «virtually all biotinylated subunits are non-globular and contain many biotinylation». I am afraid that the statement is not strong enough to support the hypothesis and a thorough analysis should be presented.

To really corroborate the hypothesis, the authors must provide data indicating significant correlation between biotinylation observed in the subunits of the 80S ribosomal complex and IDRs of such proteins (predicted or observed).

Just as we have tested for the significant over-representation of biotins in IDRs globally, we have tested it for the subset of proteins that forms the ribosome (using binomial tests). Furthermore, we have tested this separately for cytosolic ribosomal proteins as well as mitochondrial ribosomal proteins.

In the case of cytosolic ribosomal proteins, only the DiDBIT study had substantial numbers of biotin locations to perform significance tests on. Irrespective of IDR prediction algorithms, there are significantly more biotins in the IDRs of cytosolic ribosomal proteins than expected.

In the case of mitochondrial ribosomal proteins in the DiDBIT study, the number of biotins in IDRs was significantly more than expected only in the case of those IDRs predicted by the VSL2b algorithm (short IDRs).

Also, as the authors state, «Many of these (proteins) are inaccessible to water or larger molecules such as biotin in the fully assembled 80S ribosomal complex as they are contacting ribosomal RNA». This leads to the statement in the 'Discussion' that the biotinylation happens before or during assembling of the 80S ribosomal complex. Can the authors expand on the biological relevance/significance of this? Has previously ever reported in the literature that post-translational modification of the subunits is necessary for the 80S ribosomal complex?

This is a very interesting point. We list large numbers or previously reported PTMs in ribosomal subunits in table S1.2. Our understanding is that the precise biological roles (e.g. for 80S assembly) of most of ribosomal PTMs are still largely unknown. In addition to known differential regulation of subunit stoichiometry of cellular ribosomes, dynamics in ribosomal PTMs might offer additional layers of tight regulation of translation. We have included a recent reference that comprehensively analyses multiple naturally occurring types of PTMs that can be observed in ribosomal proteins (line 392-394).

REVIEWERS' COMMENTS:

Reviewer #1 (Remarks to the Author):

In this revised manuscript, the authors provided an explanation to the major concern that the DidBiT dataset did not show clear preference to the IDR. The new experimental data is welcomed. Meanwhile, the concentration of biotin reagents should be compared across the dataset. The treatment time can be compensated with appropriate concentrations.

Figure 3A show the relation between fraction of IDR for different number of biotin sites. The absolute of number of biotin sites should be correlated to the Length (or number of IDR residues), not the fraction. The authors can consider to do a similar analysis: either changing the y-axis to number of IDR residues or changing X-axis to fraction of biotin (number of biotin sites/total number of residues).

A few minor issues for authors to consider, hoping to help with the improvement of the presentation:

1. typo or grammar: Line 199: a immunofluorescence ; Line 226: might are more biotinylated
2. Figure 2 A(i-iv) cannot be read (higher resolution is required, or consider re-arranging them in two rows), hard to compare to 2A-v. Similar legends of 2B should be placed to 2A to help reading.
3. Figure 2B (i) IDR (orange) cannot be identified from Protter representation (is it all over the protein?). If it is fully disordered, then it is not necessary to color it (or only color the residues that are NOT disordered).

Reviewer #2 (Remarks to the Author):

Proximity based labeling is emerging as a powerful means to assess the spatial interactome in vivo. In this manuscript, the authors attempt to determine whether different labeling techniques such as BioID or APEX selectively label, and therefore identify, proteins with Intrinsically Disordered Regions (IDRs). By comparing four different proteome wide approaches with these various techniques, using in silico analyses only, they are able to determine both at the specific protein level and proteome wide, that labeling occurs more commonly at IDR's and that this is a feature present across all methodologies. These findings are exciting and important for biologists using these techniques to better understand the strengths and limitations of proximity based labeling. However, the original manuscript suffers from insufficient data for each methodology to be able support their conclusions (single study each, different localizations for each protein), and a lack of discussion of the unique attributes for each approach.

In the revised version of this manuscript, the authors have addressed these limitations by performing additional replication data themselves. Additionally, they have thoroughly edited the manuscript to include a discussion of the unique attributes of each approach. With these edits, the revised version merits publication in its current form.

Reviewer #3 (Remarks to the Author):

I am overall very satisfied by the amendments and the additional clarifications provided by Minde et al regarding the research paper entitled 'Biotin proximity tagging favours unfolded proteins'. Particularly, I believe that the manuscript greatly benefited from the addition of the experimental section on the

biotin painting of the bacterial 70S ribosome subunits. The mass spectrometry analysis provides a good confirmation of the computational data, giving even more credibility to the study.

However, I suggest few additional very minor amendments.

1. Fig 1A cross-reference is now missing from the main text.
2. Lines 375: reference is required.
3. Lines 376-378: "Additionally, it could indicate that elongated shapes of ribosomal proteins that show unusually large interfaces and therefore are likely to have low intrinsic stabilities upon dissociation from partner proteins and ribosomal RNA". I am not sure this sentence is clearly explained, probably there is a typo that makes it unclear. I believe the correct sentence would be "Additionally, it could indicate that elongated shapes of ribosomal proteins that show unusually large interfaces are likely to have low intrinsic stabilities upon dissociation from partner proteins and ribosomal RNA".
4. Line 386. There is a typo, "the" before ribosome.
5. Table S3.1. A legend explaining the colour-codes would be very beneficial to the reader.
6. In my previous review of the manuscript, I requested further clarification on the non-specificity of the biotinylation in the four studies analysed. The authors pointed out that "the differences in biotinylation sites between studies is likely due to multi-localised proteins changing conformation and/or protein-protein interactions in different subcellular niches". I believe this should be added in the manuscript for more clarity, either at the end of the section "Illustrative examples of proteins that are biotinylated across all four independent studies" or in the caption of figure 2.
7. In my previous review of the manuscript, I requested further clarification on the correlation biotinylation on proteins' IDRs and other PTMs in the same species. The authors diligently included Table S1.8 to answer this point and provided a figure in the rebuttal showing that there is a significant correlation between IDRs and Phosphorylation and Sumoylation but not with Ubiquitination or Acetylation. I believe this figure could be included in the supplementary material, because of its efficacious visual effect.

Overall, I recommend the publication of this paper upon these minor changes and I would like to thank the authors for their contribution on the biological function of PTMs.

REVIEWERS' COMMENTS:

Reviewer #1 (Remarks to the Author):

In this revised manuscript, the authors provided an explanation to the major concern that the DidBiT dataset did not show clear preference to the IDR. The new experimental data is welcomed. Meanwhile, the concentration of biotin reagents should be compared across the dataset. The treatment time can be compensated with appropriate concentrations.

Figure 3A show the relation between fraction of IDR for different number of biotin sites. The absolute of number of biotin sites should be correlated to the Length (or number of IDR residues), not the fraction. The authors can consider to do a similar analysis: either changing the y-axis to number of IDR residues or changing X-axis to fraction of biotin (number of biotin sites/total number of residues).

- We have amended the plot to show the length of IDR across proteins (on a log₁₀ scale for easier visualisation) vs number of biotins.

A few minor issues for authors to consider, hoping to help with the improvement of the presentation:

1. typo or grammar: Line 199: a immunofluorescence; Line 226: might are more biotinylated
 - Amended and now on line 197
 - Amended and now on line 225
2. Figure 2 A(i-iv) cannot be read (higher resolution is required, or consider re-arranging them in two rows), hard to compare to 2A-v. Similar legends of 2B should be placed to 2A to help reading.
 - We have provided higher resolution figures as suggested.
 - We have added a note in the legend to say 2B and 2A share a legend
3. Figure 2B (i) IDR (orange) cannot be identified from Protter representation (is it all over the protein?). If it is fully disordered, then it is not necessary to color it (or only color the residues that are NOT disordered).
 - Yes, it is almost entirely disordered except for the little stretch of white circles at the N-terminus.
 - Colouring just those residues that are not disordered changes the colour scheme which we have used to represent all Protter images that we have generated that are included in the links in Supplementary Data file 1, S1.3 and we want to keep illustrations consistent. Highlighting IDRs (instead of their absence) is also more directly related to the research question of this study.

Reviewer #2 (Remarks to the Author):

Proximity based labeling is emerging as a powerful means to assess the spatial interactome in vivo. In this manuscript, the authors attempt to determine whether different labeling techniques such as BioID or APEX selectively label, and therefore identify, proteins with Intrinsically Disordered Regions (IDRs). By comparing four different proteome wide approaches with these various techniques, using in silico analyses only, they are able to determine both at the specific protein level and proteome wide, that labeling occurs more

commonly at IDR's and that this is a feature present across all methodologies. These findings are exciting and important for biologists using these techniques to better understand the strengths and limitations of proximity based labeling. However, the original manuscript suffers from insufficient data for each methodology to be able support their conclusions (single study each, different localizations for each protein), and a lack of discussion of the unique attributes for each approach.

In the revised version of this manuscript, the authors have addressed these limitations by performing additional replication data themselves. Additionally, they have thoroughly edited the manuscript to include a discussion of the unique attributes of each approach. With these edits, the revised version merits publication in its current form.

Reviewer #3 (Remarks to the Author):

I am overall very satisfied by the amendments and the additional clarifications provided by Minde et al regarding the research paper entitled 'Biotin proximity tagging favours unfolded proteins'. Particularly, I believe that the manuscript greatly benefited from the addition of the experimental section on the biotin painting of the bacterial 70S ribosome subunits. The mass spectrometry analysis provides a good confirmation of the computational data, giving even more credibility to the study.

However, I suggest few additional very minor amendments.

1. Fig 1A cross-reference is now missing from the main text.
 - Added to line 105
2. Lines 375: reference is required.
 - Amended to say Supplementary Figure 4 on line 377
3. Lines 376-378: "Additionally, it could indicate that elongated shapes of ribosomal proteins that show unusually large interfaces and therefore are likely to have low intrinsic stabilities upon dissociation from partner proteins and ribosomal RNA". I am not sure this sentence is clearly explained, probably there is a typo that makes is unclear. I believe the correct sentence would be "Additionally, it could indicate that elongated shapes of ribosomal proteins that show unusually large interfaces are likely to have low intrinsic stabilities upon dissociation from partner proteins and ribosomal RNA".
 - Amended and in lines 383-384
4. Line 386. There is a typo, "the" before ribosome.
 - Amended and now on line 393
5. Table S3.1. A legend explaining the colour-codes would be very beneficial to the reader.
 - We agree and have added this to the 'Index' sheet of Supplementary Data file 3.
6. In my previous review of the manuscript, I requested further clarification on the non-specificity of the biotinylation in the four studies analysed. The authors pointed out that "the differences in biotinylation sites between studies is likely due to multi-localised proteins changing conformation and/or protein-protein interactions in different subcellular niches". I believe this should be added in the manuscript for more clarity, either at the end of the section "Illustrative examples of proteins that are biotinylated across all four independent studies" or in the caption of figure 2.
 - We agree and have added it to the end of page 6, lines 252-254.

7. In my previous review of the manuscript, I requested further clarification on the correlation biotinylation on proteins' IDRs and other PTMs in the same species. The authors diligently included Table S1.8 to answer this point and provided a figure in the rebuttal showing that there is a significant correlation between IDRs and Phosphorylation and Sumoylation but not with Ubiquitination or Acetylation. I believe this figure could be included in the supplementary material, because of its efficacious visual effect.

- We have now added this as supplementary figure S8 at the end of the paper and made reference to it in the discussion section.

Overall, I recommend the publication of this paper upon these minor changes and I would like to thank the authors for their contribution on the biological function of PTMs.

We are very grateful to all the reviewers for their positive and encouraging feedback.